# Clathrin light chains CLCa and CLCb have non-redundant roles in epithelial lumen formation

Yu Chen[1,2],*, Kit Briant[1,2],*, Marine D Camus[1,2], Frances M Brodsky[1,2]

To identify functional differences between vertebrate clathrin light chains (CLCa or CLCb), phenotypes of mice lacking genes encoding either isoform were characterised. Mice without CLCa displayed 50% neonatal mortality, reduced body weight, reduced fertility, and ~40% of aged females developed uterine pyometra. Mice lacking CLCb displayed a less severe weight reduction phenotype compared with those lacking CLCa and had no survival or reproductive system defects. Analysis of female mice lacking CLCa that developed pyometra revealed ectopic expression of epithelial differentiation markers (FOXA2 and K14) and a reduced number of endometrial glands, indicating defects in the lumenal epithelium. Defects in lumen formation and polarity of epithelial cysts derived from uterine or gut cell lines were also observed when either CLCa or CLCb were depleted, with more severe effects from CLCa depletion. In cysts, the CLC isoforms had different distributions relative to each other, although they converge in tissue. Together, these findings suggest differential and cooperative roles for CLC isoforms in epithelial lumen formation, with a dominant function for CLCa.

## Introduction

Clathrin-mediated membrane traffic is critical for a range of biological processes including tissue development, neurotransmission, metabolism, and immunity (1, 2, 3). Clathrin-mediated endocytosis and recycling from endosomes are responsible for regulating plasma membrane levels of numerous receptors and transporters, whereas clathrin-mediated transport at intracellular membranes influences formation of lysosomes, secretory granules, and specialised organelles (2, 4). Adaptor recruitment to specific membranes leads to localised clathrin self-assembly, deforming the membrane, and selectively capturing cargo into clathrin-coated vesicles for ongoing transport to target destinations (1, 5). The major form of clathrin in all eukaryotes is a triskelion-shaped trimer of three identical clathrin heavy chain

subunits (CHC17 in humans) with three associated clathrin light chains (CLCs). In some vertebrate species (present in humans, absent from rodents and ruminants), there is a second clathrin formed by CHC22 that does not bind CLCs, which is muscle-enriched and mediates specialised membrane traffic of the GLUT4 glucose transporter (1, 2). In all vertebrates, the CLCs are obligate subunits for the ubiquitous CHC17 clathrin and are encoded by two genes that undergo tissue-specific splicing, respectively, producing CLCa and CLCb proteins, each of which has a ubiquitously expressed or neuron-specific splice variant (the latter denoted by nCLCa and nCLCb) (6, 7, 8). CLCa and CLCb are ~60% identical in protein sequence with their differences and their tissue-specific expression levels and splicing patterns highly conserved across vertebrates (6, 9). These characteristics indicate evolutionary pressure to maintain the different CLC isoforms, suggesting the CLCs perform non-redundant and tissue-specific physiological functions. Here, we investigate isoform-specific CLC functions in mice following homozygous deletion of CLC-encoding genes (*Clta* and *Cltb*).

Since their identification as components of the clathrin triskelion in 1980 and appreciation of their diversity in vertebrates (10, 11), it has been challenging to discover differential roles for the CLCs. The only completely shared domain between CLCa and CLCb is a 22 amino acid consensus sequence (about 10% total length) close to their N-termini that is responsible for binding huntingtin-interacting proteins (HIP1 and HIP1R), with a homologous sequence binding the HIP-related Sla2p in yeast (12, 13, 14). The three C-terminal residues of the consensus sequence are essential for CLC's role in modulating the pH dependence of clathrin self-assembly (15). Other than the consensus sequence, the N-terminal thirds of CLCa and CLCb are divergent in length and sequence, although they are more similar in the central CHC-binding domain (66% identity) and C-terminal third (73% identity) with neuronal splicing inserts at equivalent positions (7, 8). nCLCa includes 30 amino acids encoded by two exons, and nCLCb includes 18 amino acids, encoded by one exon, with homology to the first 18 residues of the nCLCa insert (7). Functional studies of CLC contribution to clathrin pathways in tissue culture systems have revealed roles in

---

[1]Department of Structural and Molecular Biology, Division of Biosciences, University College London, London, UK    [2]Institute of Structural and Molecular Biology, Birkbeck and University College London, London, UK

Correspondence: f.brodsky@ucl.ac.uk
*Yu Chen and Kit Briant contributed equally to this work

---

G-protein–coupled receptor (GPCR) uptake, focal adhesion formation, cell migration, and invadopodia formation (16, 17, 18, 19, 20). In only a few of these, isoform-specific properties of CLCs have been identified. The epithelial splice variant of myosin VI was shown to interact with clathrin, specifically through CLCa, with consequences for clathrin-mediated endocytosis at the actin-rich apical membrane of epithelial cysts (21). A preferential role for CLCa was identified in cell spreading and migration (19), whereas isoform-specific CLCb phosphorylation was shown to influence clathrin dynamics and GPCR uptake (16). In complementary in vitro studies, reconstitution of clathrin with different CLC isoforms showed isoform-specific effects on biophysical properties of the clathrin lattice (22). Thus, how CLCs diversify clathrin function to meet specialised, tissue-specific needs in vivo remains a key question in the field. To this end, we generated homozygous mice lacking either *Clta* or *Cltb*. In these knock-out (KO) animals, we previously characterised a role for CLCa in membrane traffic controlling antibody isotype switching in B lymphocytes, which express mainly CLCa (9). We also found that depletion of nCLCa or nCLCb had opposing effects on synaptic vesicle generation and synaptic transmission (22). Notably, CLCa loss reduced synaptic vesicle formation and transmission, whereas both were increased in CLCb-depleted mice (22). These neuronal phenotypes suggested that CLCa performs more of a housekeeping role for clathrin, whereas CLCb acts to attenuate or regulate CLCa function, perhaps through its natural competition with CLCa for CHC binding (23), such that a balance of the two CLC isoforms is required for normal function.

Here, we further characterised the CLCa and CLCb KO mice to elucidate the physiological roles of CLCa and CLCb. We found that CLCa KO mice have much stronger phenotypes than CLCb. Approximately half of CLCa KO mice die within 3 d of birth. Surviving CLCa KO mice have reduced bodyweight and impaired fertility. In contrast, CLCb KO mice show very mild phenotypes, with no mortality or infertility associated with this genotype, and only a small reduction in body weight. Furthermore, we observed that loss of CLCa, but not CLCb, in female mice results in a change in endometrial epithelial cell identity and the development of uterine inflammation, indicating a critical isoform-specific role for CLCa in these cells. When analysed in vitro, we found that acute loss of either CLCa or CLCb was sufficient to prevent the generation of apico-basal polarity in 3D-cyst models of epithelial cells. Together, these results further indicate a functional dominance for CLCa in vivo and add support to the concept that CLCa plays a housekeeping role in clathrin function that is regulated by competition from CLCb, so that they operate in tandem to modulate clathrin function in epithelial lumen formation.

# Results

### Loss of CLCa reduces post-natal survival and body weight

We have previously shown that CLCa KO and CLCb KO mice have phenotypes in immune cells and neurons (9, 22). To further characterise physiological functions of CLCs, heterozygous mice were crossed ($Clta^{+/-}$ × $Clta^{+/-}$ and $Cltb^{+/-}$ × $Cltb^{+/-}$) and their

offspring compared. When weaned (3 wk of age), only 9.7% of the $Clta^{+/-}$ cross offspring were homozygous for CLCa KO, significantly lower than the expected 25% ($P = 0.0185$), indicating that loss of CLCa reduces survival at the pre- or neo-natal stage (Fig 1A). In contrast, 23.7% of $Cltb^{+/-}$ cross offspring were homozygous for CLCb KO, close to the expected 25% (Fig 1A). Thus, loss of CLCa but not loss of CLCb reduces survival. To identify the developmental stage at which survival is affected, CLCa KO genotypes were analysed pre- and post-natally. At the late embryonic (E18.5) and early post-natal stage (PN1), we observed 30.1% and 27.2% homozygous CLCa KO ($Clta^{-/-}$), respectively (Fig 1B), indicating that loss of CLCa does not affect pre-natal survival. At post-natal day 3 (PN3), significantly fewer than expected mice were homozygous CLCa KO (<10%), which remained the case at PN7 and at 4-wk old (Fig 1B) and beyond (data not shown), indicating survival rates are unaffected by the loss of CLCa after PN3. Thus, CLCa has a critical role in post-natal mouse development and/or survival behaviour, such as response to critical feeding cues, within the first 3 d of birth.

The body weight of surviving CLCa KO and CLCb KO pups at PN1 was not significantly different from that of their respective WT and heterozygous littermates (Fig 1C). However, at PN7, both CLCa KO and CLCb KO pups had a significantly lower body weight than WT pups, respectively, 21.52% and 19.1% (Fig 1D). CLCa KO adult male and female mice (over 8 wk of age) maintained a significantly lower body weight than WT littermates, with male mice more affected (20.8% reduction in male CLCa KO body weight at 17–24 wk of age compared with a 12.8% weight reduction for female CLCa KO mice, Fig 1E). No significant difference in the body weight of CLCb KO mice compared with WT littermates above 8 wk of age was observed (Fig 1E).

To establish whether the observed mortality and body weight phenotypes were associated with gross morphological changes in the organs of the CLCa KO and CLCb KO mice, tissue slices from mice of each genotype were analysed by hematoxylin and eosin staining. No abnormalities in the structure or morphology of small intestine, skeletal muscle, kidney, and heart tissues from surviving CLCa KO adult mice or CLCb KO adult mice compared with WT adult mice (aged between 5 and 9 mo) were observed (Fig S1).

### Loss of CLCa, but not CLCb, reduces fertility in mice

During routine maintenance of CLCa KO and CLCb KO mouse lines, we observed fewer than expected litters when CLCa KO mice were used for breeding. To analyse this further, breeding cages were set up with combinations of WT, heterozygous, and KO animals derived from heterozygous crosses ($Clta^{+/-}$ × $Clta^{+/-}$ or $Cltb^{+/-}$ × $Cltb^{+/-}$), and the number of litters born within 54 d recorded (Fig 2A). Whereas WT breeding pairs produced 1–2 litters during this period, CLCa KO × CLCa KO breeding crosses produced no offspring at all, suggesting loss of CLCa affects fertility. In contrast, loss of CLCb did not reduce the number of litters born (Fig 2A). Crosses of heterozygous $Clta^{+/-}$ mice with CLCa KO animals also generated a significantly reduced number of litters born within 54 d (Fig 2A). However, breeding WT with CLCa KO animals in either male/female combination produced litter numbers similar to WT breedings during the 54 d period (Fig 2A). A separate analysis over a 4 mo (120 d) period of breeding

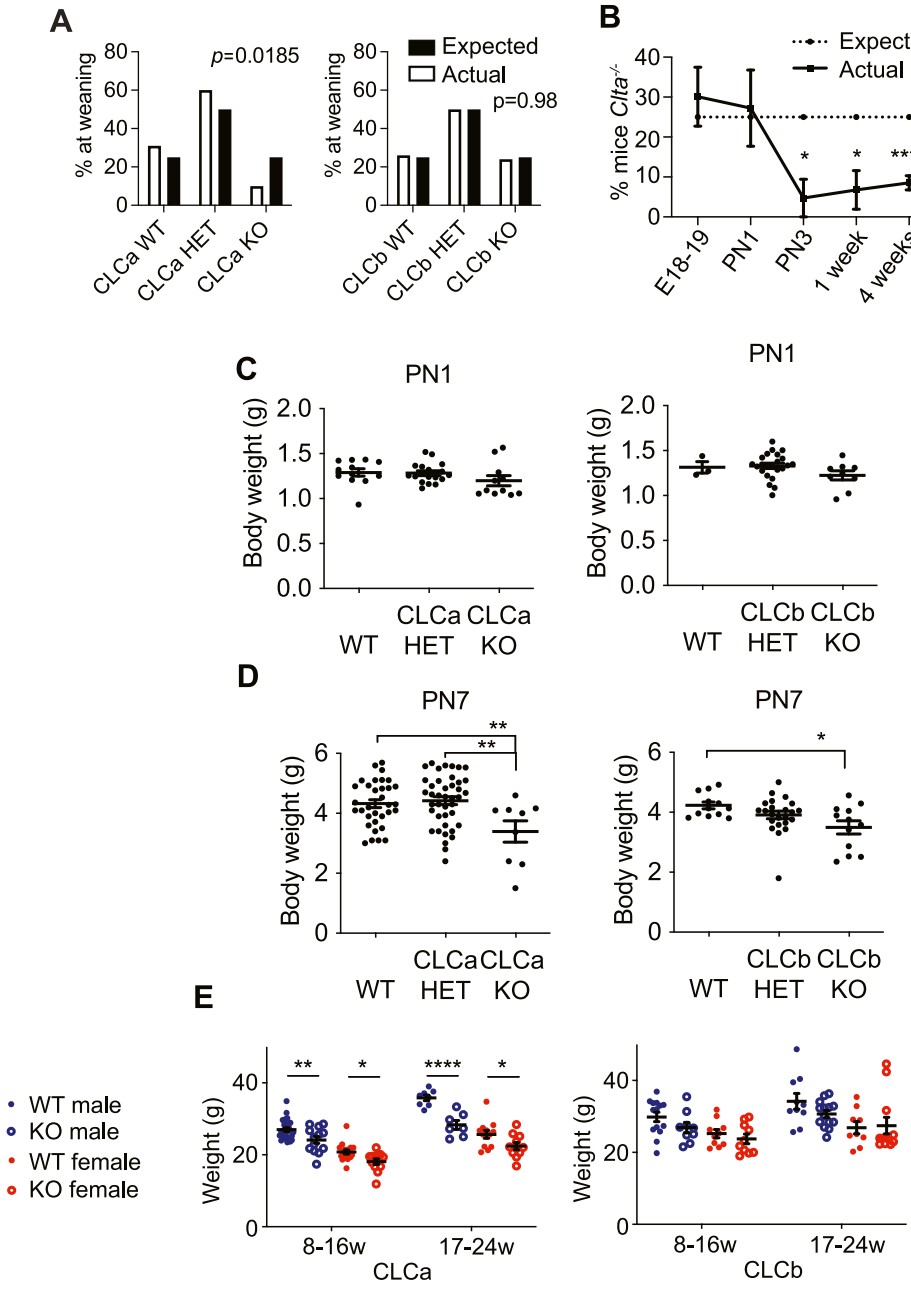

**Figure 1. Survival and body weight of mice lacking genes encoding CLCa and CLCb.**
**(A)** Breeding cages with $Clta^{+/-} \times Clta^{+/-}$ or $Cltb^{+/-} \times Cltb^{+/-}$ parental genotypes were established and the genotype of offspring analysed at weaning (3 wk old). Expected and observed percentages for the genotypes $Clta^{+/+}$ (CLCa WT), $Clta^{+/-}$ (CLCa HET), $Clta^{-/-}$ (CLCa KO), $Cltb^{+/+}$ (CLCb WT), $Cltb^{+/-}$ (CLCb HET), $Cltb^{-/-}$ (CLCb KO) are shown. Number of mice analysed: CLCa WT = 303, CLCa HET = 589, CLCa KO = 96, total = 988; CLCb WT = 181, CLCb HET = 344, CLCb KO = 163, total = 688. P-values generated by Chi Square analysis comparing the observed genotype percentage of the mice with the expected Mendelian ratios. **(B)** Genotype analysis of CLCa KO mice at developmental stages following $Clta^{+/-} \times Clta^{+/-}$ breeding crosses. The percentage of $Clta^{-/-}$ mice per litter at E18.5 (n = 8 litters), post-natal day 1 (PN1, n = 5 litters), PN3 (n = 7 litters), 1 wk-PN7 (n = 5 litters), and 4 wk old (n = 52 litters) is shown. *P < 0.05, ****P < 0.0001, Fisher's exact test. **(C, D)** Body weight of mice from $Clta^{+/-} \times Clta^{+/-}$ or $Cltb^{+/-} \times Cltb^{+/-}$ breeding crosses at PN1 (C) and PN7 (D) is shown in grams (g) for the indicated genotypes. Number of mice measured: $Clta^{+/-} \times Clta^{+/-}$ breeding crosses (PN1 = 41; PN7 = 83) and $Cltb^{+/-} \times Cltb^{+/-}$ breeding crosses (PN1 = 35; PN7 = 47). *P < 0.05, **P < 0.01, one-way ANOVA test, with Holm-Sidak's multiple comparison. **(E)** Body weight of adult male and female mice (aged 8–16 or 17–24 wk) for WT or homozygous KO genotypes from $Clta^{+/-} \times Clta^{+/-}$ (CLCa) or $Cltb^{+/-} \times Cltb^{+/-}$ (CLCb) breeding crosses. Number of mice measured from $Clta^{+/-} \times Clta^{+/-}$ crosses: aged 8–16 wk (CLCa WT male = 30, CLCa KO male = 13, CLCa WT female = 17, CLCa KO female = 12); aged 17–24 wk (CLCa WT male = 8, CLCa KO male = 7, CLCa WT female = 12, CLCa KO female = 11). Number of mice from $Cltb^{+/-} \times Cltb^{+/-}$ crosses: aged 8–16 wk (CLCb WT male = 14, CLCb KO male = 9, CLCb WT female = 10, CLCb KO female = 10); aged 17–24 wk (CLCb WT male = 10, CLCb KO male = 14, CLCb WT female = 9, CLCb KO female = 12). *P < 0.05, **P < 0.01, ****P < 0.0001, two-way ANOVA test, with Holm-Sidak's multiple comparison.

showed that when either CLC KO strain was bred with WT, no significant difference in the number of litters or neonates per litter was observed (Fig 2B and C). Thus, fertility was reduced between animals with no CLCa or between CLCa KO and animals (male or female) heterozygous for CLCa loss. Although this fertility defect was not observed when one mate was WT, these breeding experiments support that loss of CLCa affects fertility.

### Loss of CLCa results in pyometra and reduction in uterine glands

Consistent with loss of CLCa-affecting reproduction, we observed that 42.9% of CLCa KO female mice developed a swollen abdomen after 4 mo of age due to an enlarged uterus containing cloudy fluid (Fig 3A and B). The uterus is composed of the outer myometrium and the inner endometrium compartments. The endometrium comprises the stroma, glands surrounded by glandular epithelium and the lumen surrounded by lumenal epithelium. H&E staining revealed gross structural abnormalities within the endometrium of CLCa KO mice with enlarged uteri. The cross-sectional width of the endometrium of these mice was narrower than that in WT animals and the endometrial glandular structures were lost (Fig 3C and D). Quantification confirmed that the number of glands in CLCa KO enlarged uteri was significantly lower than the number of glands in unaffected uteri from CLCa KO, CLCb KO or WT mice (Fig 3D).

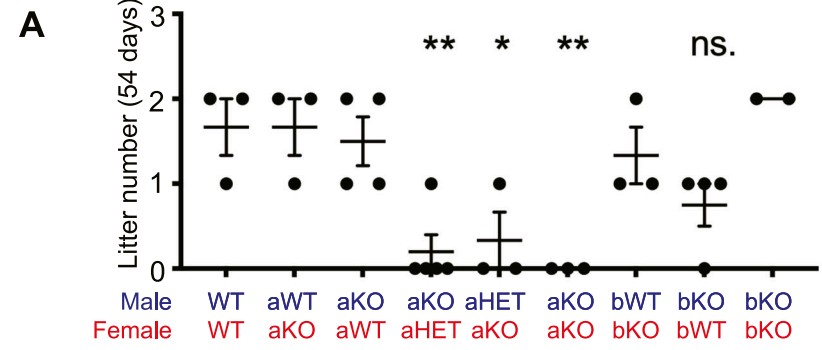

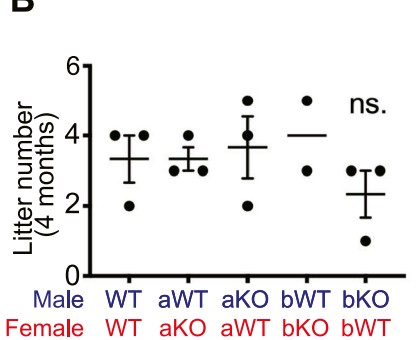

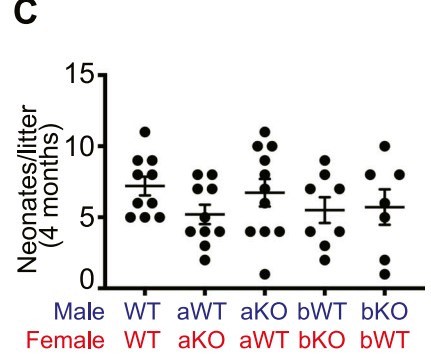

**Figure 2. Fertility of mice lacking genes encoding CLCa and CLCb.**

**(A, B)** Breeding cages containing male and female mice with the indicated genotypes $Clta^{+/+}$ (aWT), $Clta^{+/-}$ (aHET), $Clta^{-/-}$ (aKO), $Cltb^{+/+}$, (bWT), $Cltb^{+/-}$ (bHET), $Cltb^{-/-}$ (bKO) were established and the number of litters born within 54 d (A) or 4 mo (B) recorded. Each dot represents the number of litters generated by each breeding pair. Bars represent mean ± SEM, *P < 0.05, **P < 0.01, not significant (ns) compared with WT control, one-way ANOVA test, with Holm-Sidak's multiple comparison. **(C)** Breeding cages containing male and female mice with the indicated genotypes were established for 4 mo and the number of pups per litter recorded. Graph shows mean ± SEM, one-way ANOVA test, with Holm-Sidak's multiple comparison.

An enlarged uterus filled with cloudy fluid can be indicative of pyometra, a uterine infection characterised by inflammation and a "pus-filled" uterus (24). To determine whether CLCa KO mice with an enlarged uterus had developed pyometra, the inflammation status of the uteri of these mice was examined and compared with unaffected uteri of all genotypes (WT, CLCb KO, and CLCa KO). When sections of uterus were stained with antibodies against the neutrophil marker Ly6G, an indicator of inflammation (25), Ly6G-positive cells were detected in the uterine epithelial layer of CLCa KO mice with an enlarged uterus but not in unaffected animals of any genotype (Fig 3E and F). This sign of neutrophil infiltration is consistent with pyometra as the cause of the enlarged uterus in CLCa KO mice.

To further understand the role of the CLCs in the uterus, we investigated the expression levels of CLCa and CLCb in the uterus. Relative levels of CLCa versus CLCb were determined for whole tissue lysates from the uterus, spleen, and brain of WT, CLCa KO (without pyometra), and CLCb KO mice by immunoblotting using the antibody CON.1, recognising the 22 amino acid CLC consensus sequence shared between CLCa and CLCb (26). Thus, the protein levels of the different CLCs can be directly compared within a blot. In the uterus of WT mice, CLCa protein levels were marginally higher than CLCb levels, with CLCa comprising 60.5% ± 3.5% of total CLCs, similar to their relative expression in the liver and not as extreme as CLCa dominance in the spleen (9) (Fig 4A and B). More balanced levels of the two CLC isoforms were observed for nCLCa and nCLCb in the brain (Fig 4A) and CLCa versus CLCb in muscle (9). Note that the apparent dominance of nCLCb in brain tissue is due to the presence of cells with non-neuronal CLCa that co-migrates with nCLCb during SDS–PAGE (as detected by CON.1 in the CLCb KO mice). Only a slight apparent increase in CLCb level was observed in the tissues of the CLCa KO mice,

and vice versa, as noted in analysis of other tissues from these mice (9) (Fig 4A). Thus, development of pyometra correlates with CLCa loss and inability of residual or slightly elevated CLCb to support the clathrin function needed for the impaired pathway, revealing differences in the functions supported by the two CLC isoforms.

The relative localisations of CLCa and CLCb in the uterus of WT, CLCa KO, and CLCb KO mice were then determined by immunofluorescence. Both CLCa and CLCb were highly enriched in the lumenal and glandular epithelia of the uterus, with both concentrated at the apical pole of the epithelial cells (Fig 4C and D). This is consistent with a role for CLCa at apical domain of epithelia, previously suggested by the specific binding of the epithelial splice variant of myosin VI to CLCa (21) and suggests an additional role for CLCb.

### CLCa loss alters endometrial epithelial cell identity

The advent of pyometra suggests a defect in epithelial integrity, leading to inflammation. To further assess the consequences of CLCa loss on endometrial epithelia, we analysed the levels of FOXA2 expression in the uterus of WT, CLCa, and CLCb KO mice. FOXA2 is a transcription factor that regulates epithelial differentiation and development, and its loss affects gland development, uterine function, and fertility (27, 28, 29). In the mature endometrium, expression of FOXA2 is confined to glandular epithelium and absent from epithelia bordering the uterine lumen, as seen the uterine tissue from WT mice (Fig 5A and B). Ectopic expression of FOXA2 was observed in epithelial cells at the uterine lumen of CLCa KO animals with pyometra and also, to a lesser extent, in the CLCa KO mice that did not suffer from pyometra (Fig 5A and B). In contrast, loss of CLCb

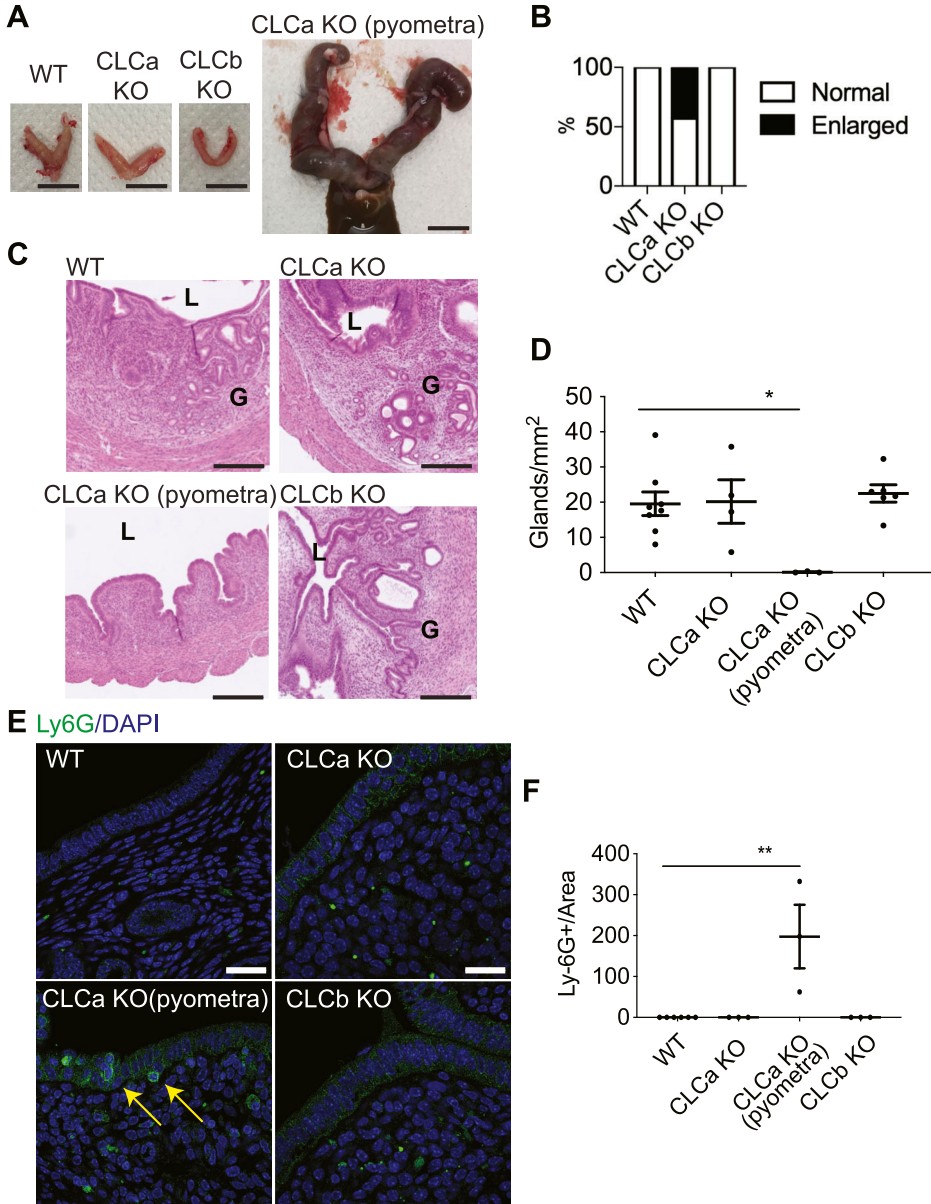

**Figure 3. CLCa KO female mice develop pyometra.**
**(A)** Images of the uterus from WT, CLCa KO (unaffected), and CLCb KO mice and a CLCa KO mouse with pyometra. Scale bar = 1 cm. **(B)** The percentage of WT, CLCa KO, and CLCb KO female mice that developed an enlarged uterus after 4 mo of age. n = 31 per genotype. **(C)** H&E staining of the uterus from WT, CLCa KO, and CLCb KO mice and a CLCa KO mouse with pyometra. Images shown are the representative cross-sectioned images of at least three mice in each genotype or condition. L = lumen, G = glands. Scale bar = 250 $\mu$m. **(D)** Number of glands per area (millimeter$^2$) of uterine tissue cross-section from WT, CLCa KO, CLCa KO with pyometra, and CLCb KO mice. Graph shows mean ± SEM. Number of mice analysed (one section per animal): WT = 8; CLCa KO (unaffected) = 4; CLCa KO (pyometra) = 3; CLCb KO = 6. *$P < 0.05$, one-way ANOVA with Holm-Sidak's multiple comparison. **(E)** Slices of uterine tissue of the indicated genotype were fixed and stained with antibody against the Ly6G neutrophil marker for inflammation. Arrows indicate Ly6G-positive cells. Nuclei were stained with DAPI (blue). Scale bar = 25 $\mu$m. **(F)** Quantification of the number of Ly6G-expressing cells per millimeter$^2$ tissue. Each dot represents the average number of Ly6G-expressing cells in endometrial tissue of one mouse. At least two confocal images were counted per mouse. Graph displays mean ± SEM; Number of mice analysed: WT = 6; CLCa KO (unaffected) = 3; CLCa KO (pyometra) = 3; CLCb KO = 3. **$P < 0.01$, one-way ANOVA test, with Holm-Sidak's multiple comparison.

did not change FOXA2 expression in the uterus, with expression seen only in the glandular epithelium (Fig 5A and B). FOXA2 has previously been shown to regulate the proliferation of endometrial epithelia (30, 31). To assess whether ectopic expression of FOXA2 in the lumenal epithelia of the uterus of CLCa KO mice altered cell proliferation, we analysed the expression of the postulated endometrial stem cell marker SRY-box transcription factor 9 (SOX9) (32) and the proliferation marker Ki67. However, the expression of both SOX9 and Ki67 in endometrial epithelial cells was unaffected by the loss of CLCa or CLCb compared with WT animals (Fig S2A–C), indicating that the ectopic expression of FOXA2 in the endometrial epithelia does not affect proliferation of these cells. The endometrial epithelium normally consists of simple columnar epithelial cells; however, overexpression of FOXA2 in the endometrium has previously been shown

to induce epithelial stratification (33). We therefore analysed the expression of keratin 14 (K14), a marker of stratified squamous epithelial cells (34). As expected, the endometrial epithelia of WT, CLCb KO, and CLCa KO mice without pyometra did not express K14 (Fig 6A and B). In contrast, K14-expressing cells were found in endometrial epithelia in uteri from 40% (two of five analysed) CLCa KO mice with pyometra (Fig 6A and B). Consistently, the cells expressing K14 had a more squamous shape (Fig 6A), indicating a change in epithelial subtype in the uterus of some of the mice with pyometra.

FOXA2 overexpression in mouse uterine tissue has previously been shown to alter expression of FOXA2 target genes (33). To test whether FOXA2 transcriptional activity was elevated in CLCa KO mice, we analysed the expression of FOXA2 target genes including *Ltf* and *Muc1* in the uterus of WT, CLCa KO, and CLCb KO mice and CLCa KO

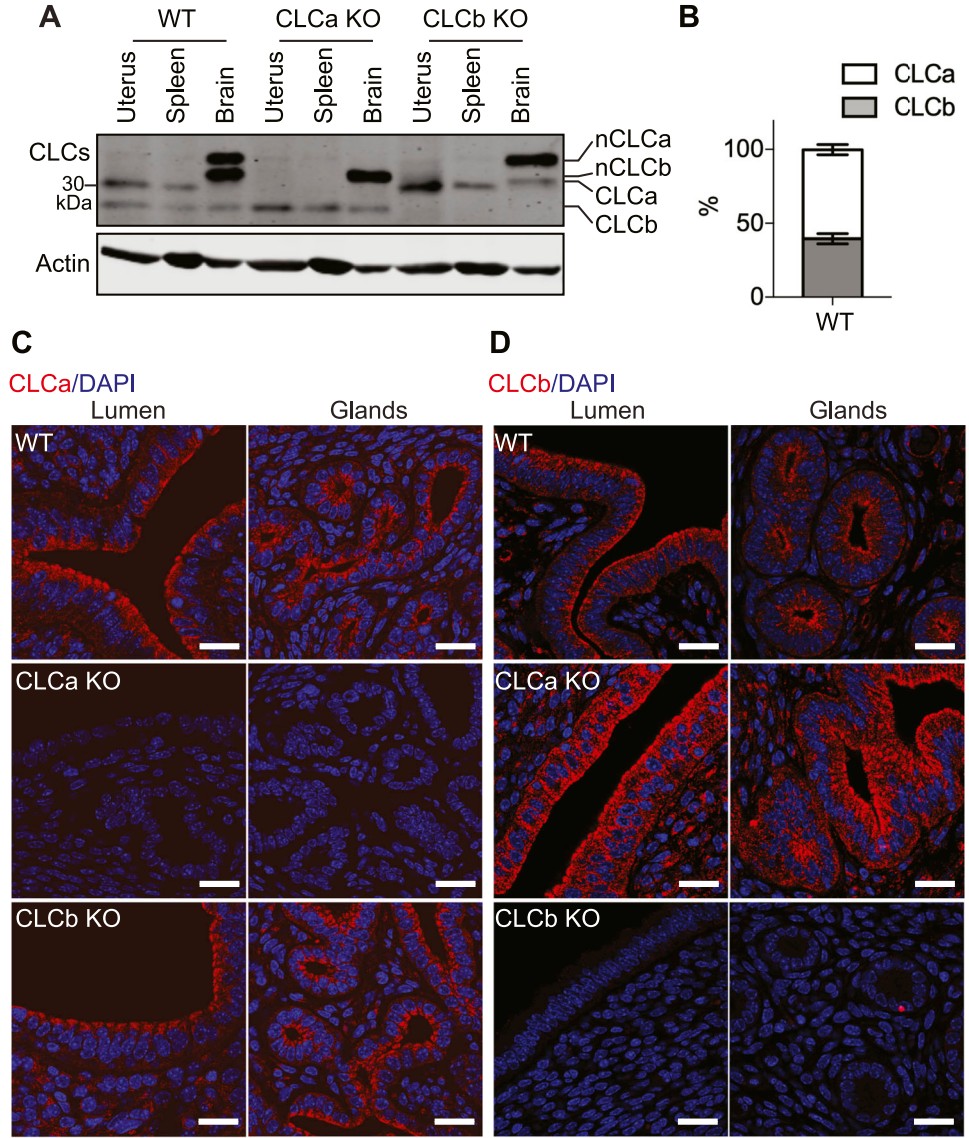

**Figure 4. Expression and localisation of CLCa and CLCb in the uterus.**
**(A)** Tissue lysates (50 µg of protein) from the uterus, spleen, and brain of WT, CLCa KO, or CLCb KO mice were analysed by SDS–PAGE and the levels of CLC isoforms compared by immunoblotting (upper panel, CLCs) using the antibody CON.1 that recognises the consensus sequence shared by CLCa and CLCb and their neuronal splice variants nCLCa and nCLCb. Migration positions of the CLC isoforms are shown right and for molecular mass marker in kilodaltons (kDa) is shown left. The lower panel shows the same samples immunoblotted for actin. **(B)** Quantification of the amount of CLCa and CLCb found in the in the uterus of WT mice by immunoblotting, shown as a percentage of the total CLC level. n = 3. **(C, D)** Slices of uterine tissue of the indicated genotype were fixed and stained with antibodies against CLCa (C) or CLCb (D), both shown in red. Nuclei were stained with DAPI (blue). Representative images of endometrial lumen and glands are shown. Scale bar = 25 µm.

mice with pyometra. Only inconsistent expression of these target genes was observed (Fig S3A), but these were whole uterine tissue samples, and not specifically epithelia. To follow-up, we then transfected a construct encoding Flag–FOXA2 into Ishikawa tissue culture cells of endometrial epithelial origin (which do not express FOXA2) to see if transcription of previously identified FOXA2 target genes (33, 35), and keratin 14, was induced. Whereas the expression of several genes (*LIF*, *LTF*, *MUC1*) remained unchanged after ectopic FOXA2 expression, expression of several other genes (*AREG*, *C3*, *IHH*, and *KRT14*) was variable. Notably, transcription of the pancreatic trypsin inhibitor *SPINK1*, which is not endogenously expressed in Ishikawa cells, was significantly increased with Flag–FOXA2 expression (Fig S3B–E). Thus, although inconsistent, ectopic FOXA2 expression did alter transcription of at least a subset of target genes in uterine epithelia, as reported elsewhere (33, 35).

The structure and integrity of the endometrial epithelia at the lumen in the CLC KO mice was further assessed by immunofluorescent staining of epithelial adherens junction component E-cadherin and the apical tight junction protein ZO-1. WT, CLCa KO with and without pyometra, and CLCb KO mice all displayed normal distribution of both E-cadherin and ZO-1, with ZO-1 distributed at the apical surface of the endometrial epithelium and E-cadherin distributed at the basolateral membrane (Fig S4A and B). These findings suggest that epithelial junctions are formed normally in all CLC KO mice. Thus, at the resolution detectable by confocal microscopy, loss of CLCa or CLCb does not alter the organisation or polarity of the endometrial epithelia. The expression and distribution of mucin 1 (MUC1) was also examined because MUC1 is found at the apical surface of uterine epithelia and acts as a barrier against microbial invasion (36). Loss of this barrier could therefore lead to infection and pyometra. However, a layer of MUC1 was observed at the apical surface of endometrial epithelia of the uterine lumen from WT, CLCa KO, and CLCb KO animals (Fig S4C), suggesting that a loss of apical MUC1 is not the cause of

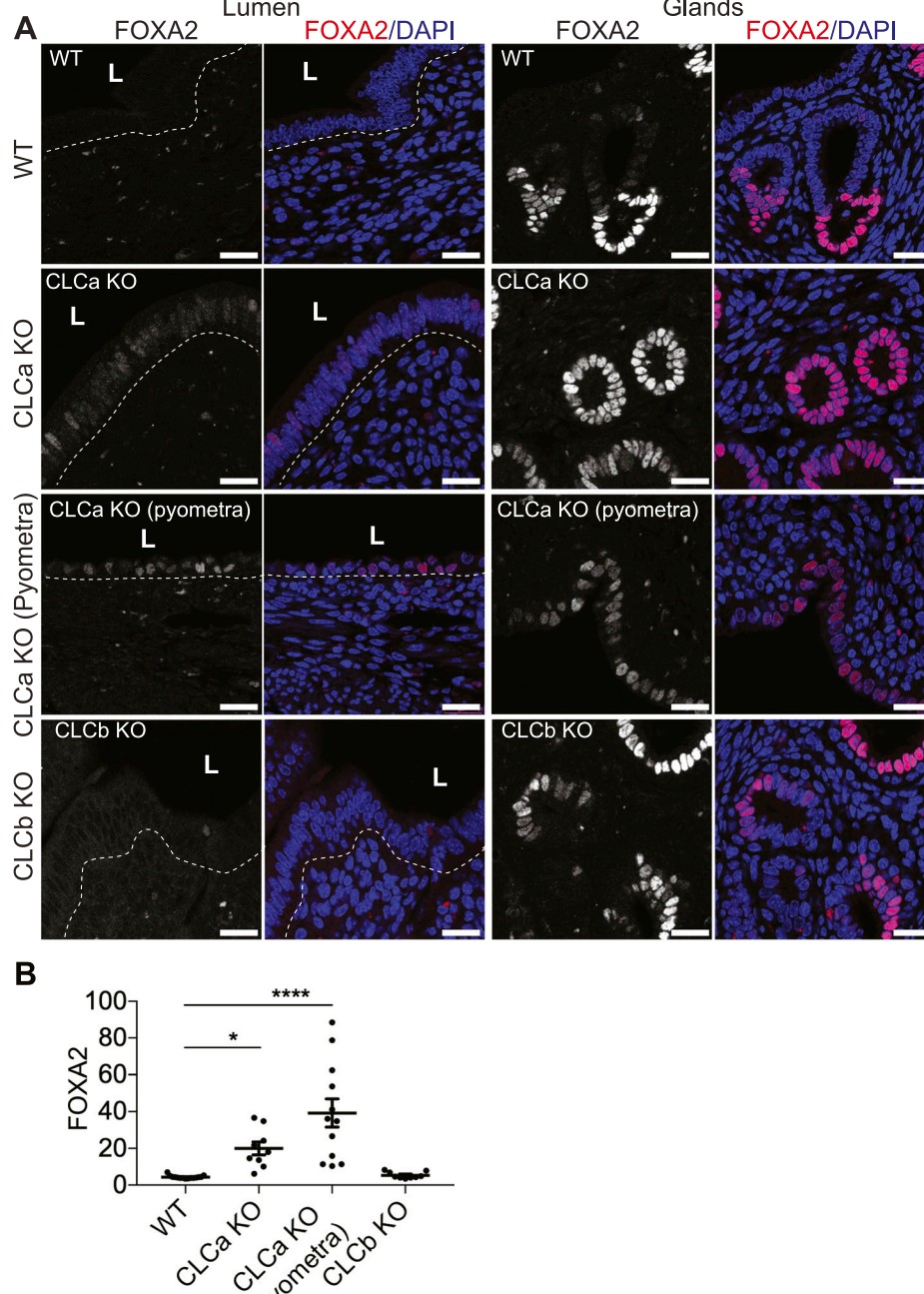

**A**

Lumen | Glands

FOXA2 FOXA2/DAPI FOXA2 FOXA2/DAPI

WT

CLCa KO

CLCa KO (Pyometra)

CLCb KO

**B**

FOXA2

WT · CLCa KO · CLCa KO (pyometra) · CLCb KO

**Figure 5. FOXA2 is ectopically expressed in lumenal cells of the endometrium epithelium in CLCa KO mice.**
**(A)** Immunostaining for FOXA2 in endometrium from WT, CLCa KO (± pyometra), or CLCb KO mice. Slices of uterine tissue of the indicated genotype and phenotype were fixed and stained with antibodies against FOXA2 (pink in merge). Nuclei were stained with DAPI (blue). The dotted line shows the boundary between the lumenal epithelium and the rest of the endometrium and the position of the lumen (L) is indicated. Representative images of endometrial lumen and glands are shown. Scale bar = 25 $\mu$m. **(B)** Mean fluorescence intensity for FOXA2 in cells of the lumenal epithelium. Each dot represents the average mean fluorescence intensity of the masked nuclear region of all lumenal cells in one confocal image as assessed by Image J. Three images were analysed per animal. Graph displays mean ± SEM; number of mice analysed: WT = 5; CLCa KO (unaffected) = 3; CLCa KO (pyometra) = 4; CLCb KO = 3. *$P < 0.05$, ****$P < 0.0001$ on one-way ANOVA test, with Holm-Sidak's multiple comparison.

pyometra in CLCa KO animals. Thus the pyometra phenotype may be linked to regional disruption of the epithelium by aberrant conversion to stratified epithelial cells, this differentiation defect being the only defect detected in our analysis.

**Loss of CLCa or CLCb disrupts epithelial cyst polarity in vitro**

Analysis of uterine tissue suggested that CLCs play a role in epithelial cell differentiation. However, despite the high level of expression of CLCa and CLCb in these cells, loss of either CLCa or CLCb did not result in observable organisational defects in the epithelial

layer of the uterine lumen of CLCa KO or CLCb KO mice. The mice used in this study are a constitutive knock-out model, and it is possible that independent compensatory pathways or some substitute functions shared by CLCa and CLCb may have been activated and are sufficient to obscure individual functions of each CLC isoform in these cells. To address this possibility, we assessed the ability of cultured epithelial cell lines to form 3-dimensional cysts in vitro upon acute knockdown of either CLC or clathrin heavy chain (CHC17). Ishikawa cells (human endometrial adenocarcinoma cells) and Caco-2 cells (human colorectal adenocarcinoma cells) were treated with control siRNA or siRNA targeting CLCa, CLCb, or CHC17

**A**

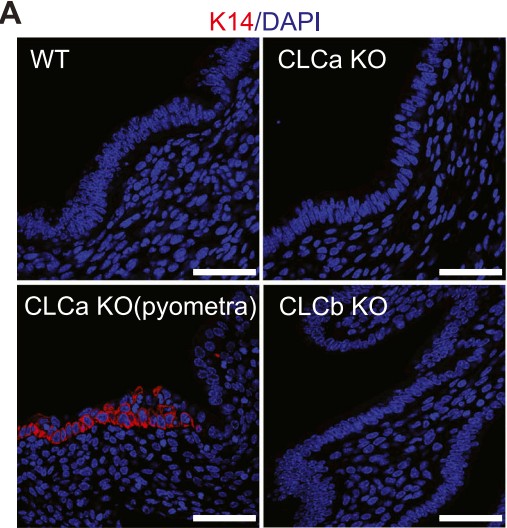

K14/DAPI

WT | CLCa KO

CLCa KO(pyometra) | CLCb KO

**B**

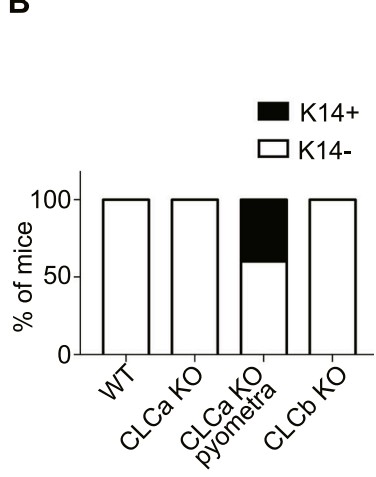

Figure 6.   **The stratified epithelial marker K14 is ectopically expressed in the endometrium of a subset of CLCa KO mice with pyometra.** **(A)** Immunostaining for K14 in the endometrium of WT, CLCa KO, or CLCb KO mice. Slices of uterine tissue of the indicated genotype were fixed and stained with antibody against K14 (red). Nuclei were stained with DAPI (blue). Scale bar = 25 $\mu$m. **(B)** Quantification of the percentage of mice for each genotype that had uterine lumenal epithelial cells expressing K14 (K14+) or did not have any uterine epithelial cells expressing K14 (K14−) as determined by immunostaining. One section per mouse. Number of mice analysed: WT = 8; CLCa KO = 4; CLCa KO (pyometra) = 5; CLCb KO = 5.

for 72 h. Treated cells were then trypsinised and seeded as single cells embedded within the extracellular matrix substrate Matrigel and grown for 4–6 d to allow cysts to form before analysis by immunofluorescence. Clathrin component depletion was confirmed by immunoblotting lysate from Ishikawa and Caco-2 cells treated with siRNA for 72 h (Fig S5B and E), and immunoblotting showed sustained depletion of clathrin components during 2D culture of Ishikawa cells for another 6 d (the period of cyst differentiation) (Fig S5B). Persistence of clathrin component depletion for both Ishikawa cells and Caco-2 cells was confirmed by immunofluorescence in 6-d cysts, with some variability in Caco-2 cysts (Fig S5A, C, and D). Ishikawa and Caco-2 cells treated with control siRNA formed symmetrical, spherical cysts with a single lumen surrounded by an F-actin-rich apical membrane (Fig 7A and B). In both Ishikawa and Caco-2 cysts, CLCa and CLCb displayed distinct distribution patterns relative to each other. In the cysts, CLCa had a more ubiquitous distribution whereas CLCb localisation was enriched towards the apical surface of the cells (Fig 7A and B). The 3D-cysts formed by Ishikawa and Caco-2 cells treated with control siRNA displayed the expected basolateral distribution of E-cadherin and ZO-1 localised tightly around the actin-rich lumen (Fig 7C). A complete loss of this apico-basal polarity was seen in Ishikawa or Caco-2 cysts depleted of CLCa or CHC17 and also for Caco-2 cysts depleted of CLCb. In all of these cases, neither E-cadherin nor ZO-1 were polarised (Fig 7C–E), and there was no symmetrical distribution of cells around a lumen (Fig 7C–E). Analysis of E-cadherin, ZO-1, and actin distribution in Ishikawa cysts depleted of CLCb showed that some apico-basal polarity was attained in these cysts, with E-cadherin showing a degree of basolateral distribution, and greater symmetry observed across the cyst than for CLCa-depleted cysts (Fig 7C). When cyst size was quantified, Caco-2 cysts were found to be reduced in size when depleted for CLCa, CLCb, or CHC17, whereas this was not observed consistently for Ishikawa cysts treated in the same way (Fig 7F and G). However, DAPI staining, which typically reveals a uniform ring of cells in control-treated cysts, showed aberrant positioning of cells in the CLC-depleted Ishikawa cysts, and we frequently observed an abnormal number of lumens within the cysts (Fig 7C). During in vitro cyst growth, lumen formation is directly linked with apico-basal polarity development and organisation, as a single lumen develops next to the forming apical membrane initiation site and subsequent pre-apical patch (37, 38). Thus, loss of apico-basal polarity leads to abnormal lumen development. Therefore, to quantify the disruption of apico-basal polarity organisation in cysts depleted of CLCa, CLCb, or CHC17 observed in Fig 7C–E, a luminogenesis assay was performed. Cells were treated with siRNAs before growing cysts for 6 d. For the final 24 h, cysts were treated with cholera toxin to expand the lumen, then fixed, and the number of lumens were scored according to the phenotypes shown for Caco-2 cysts in Fig 7H. Phenotypes were detected using phalloidin staining of apical actin rings. The apico-basal polarity of these cysts was assessed by ZO-1 and E-cadherin staining, which confirmed that loss of a single lumen detected by phalloidin coincided with disruption of these polarity markers (Fig S6). ~80% of control-treated cysts contained the expected single lumen for both Caco-2 cysts and Ishikawa cysts, which was significantly reduced in all cases by treatment with siRNA targeting CLCa, CLCb, or CHC17 (Fig 7I and J). Thus, although constitutive loss of either CLCa or CLCb did not appear to disturb the apico-basal polarity of the endometrial epithelia in the knock-out mice, acute knockdown of either CLCa or CLCb was sufficient to disturb the apico-basal polarity of 3D cysts from endometrial and colonic epithelial cells in vitro, suggesting the CLCs play an important role in epithelial cell differentiation. The more severe phenotypes seen for both cysts upon CLCa siRNA treatment compared with CLCb siRNA treatment is consistent with a dominant role for CLCa in epithelial development.

## Discussion

Mice lacking the genes encoding CLC subunits CLCa or CLCb were produced to address how CLC diversity contributes to clathrin function in tissues and the whole organism. Previous studies of

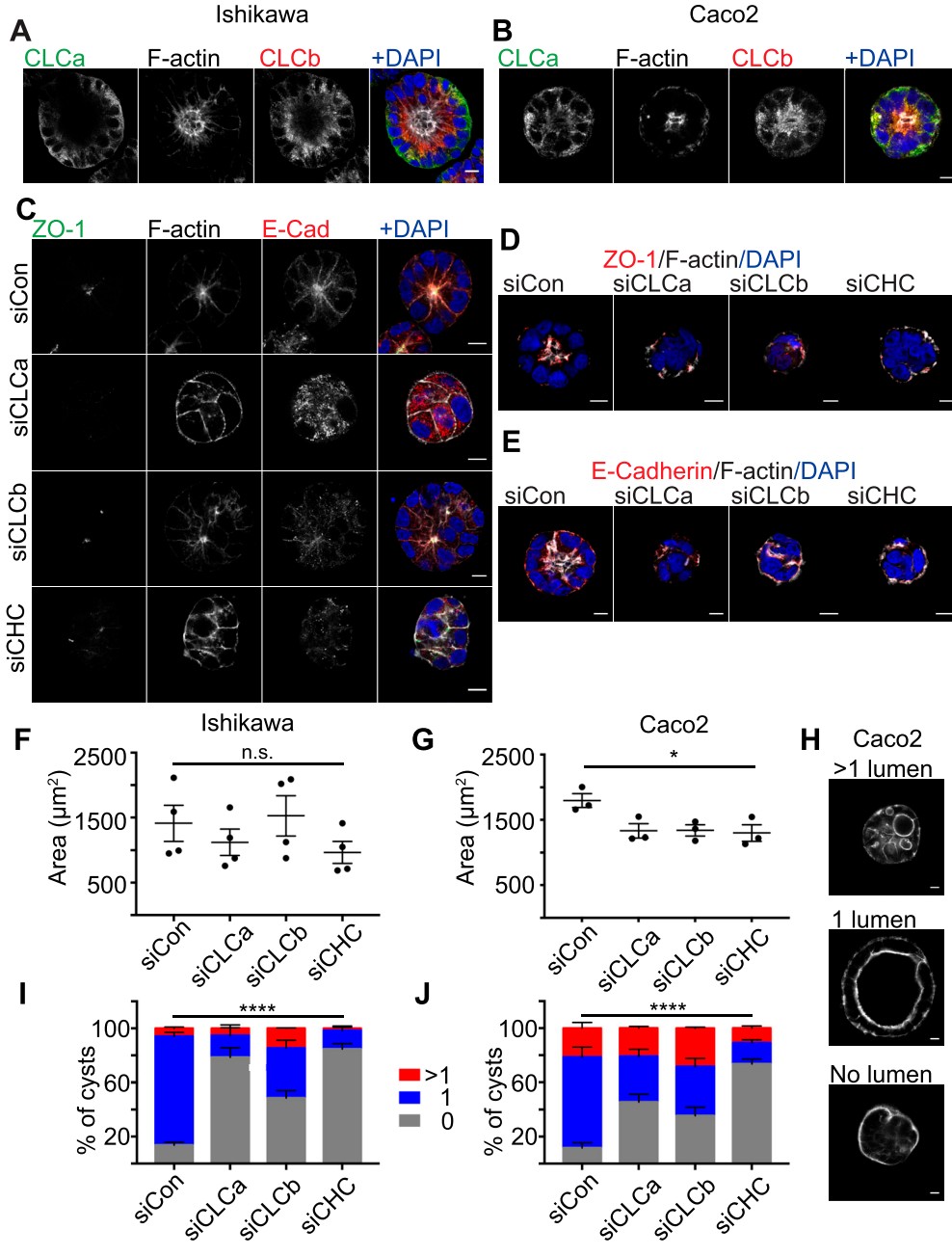

**Figure 7. Acute depletion of CLCa or CLCb in Ishikawa and Caco-2 cells disrupts epithelial cyst formation and polarisation.**
**(A, B)** Ishikawa (A) or Caco-2 (B) cells were seeded as single cells in Matrigel and grown for 6 d to allow cyst formation. Cysts were treated with cholera toxin for the final 24 h to expand the lumens for visualisation. Cysts were fixed and immunostained with antibodies against CLCa (green in merge) and CLCb (red in merge). F-actin and nuclei were stained with phalloidin (grey) and DAPI (blue), respectively. Scale bar 10 μm.
**(C, D, E)** Ishikawa (C) or Caco-2 (D, E) cells were transfected with siRNA to deplete CLCa (siCLCa), CLCb (siCLCb), or CHC17 (siCHC), or with control siRNA (siCon). 72 h after siRNA transfection, cells were trypsinised and seeded as single cells in Matrigel and grown for 4 d to allow cyst formation. Cysts were fixed and immunostained with antibodies against ZO-1 (green in (C), red in (D)) and E-cadherin (red in (C, E) merge). F-actin and nuclei were stained with phalloidin (grey) and DAPI (blue), respectively. Scale bar = 10 μm.
**(F, G)** Ishikawa (F) or Caco-2 (G) cells were transfected with siRNA to deplete CLCa (siCLCa), CLCb (siCLCb), or CHC17 (siCHC), or with control siRNA (siCon). 72 h after siRNA transfection, cells were trypsinised and seeded as single cells in Matrigel and grown for 4 d to allow cyst formation before fixation. The cross-sectional area of cysts was visualised with antibodies against E-cadherin and the F-actin stain phalloidin and measured using ImageJ software. Each dot represents the average size of over 100 cysts from a single experiment. *P = 0.0317, one-way ANOVA. Number of experiments performed: Ishikawa cysts = 4; Caco-2 cysts = 3. **(H, I, J)** Ishikawa or Caco-2 cells were transfected with siRNA to deplete CLCa (siCLCa), CLCb (siCLCb), or CHC17 (siCHC), or with control siRNA (siCon). 72 h after siRNA transfection, cells were trypsinised and seeded as single cells within Matrigel and grown for 6 d to allow cyst formation. Cysts were treated with cholera toxin for the final 24 h to expand the lumens for visualisation. Cysts were fixed and lumens visualised with the F-actin stain phalloidin. Representative images of Caco-2 cysts with multiple lumens, a single lumen or no lumen, scale bar 10 μm (H). Percentage of Ishikawa cysts with the indicated number of lumens (0, 1, >1), error bars show SEM (n = 3) (I). Percentage of Caco-2 cysts with the indicated number of lumens, error bars show SEM (n = 3) (J). ****P > 0.0001, two-way ANOVA.

these KO animals focussed on the most detectable phenotypes displayed in immune cells and neurons (9, 22). Recent findings that CLCa (and not CLCb) binds an epithelial-specific splice variant of myosin VI (21) suggested the possibility of epithelial phenotypes, which had not yet been detected in the KO mice. To search for additional phenotypes resulting from loss of each CLC isoform, general properties of the CLC KO mice including survival, fertility, and weight were assessed over time and a new phenotype, relating

to a uterine epithelial defect, was discovered and characterised. Previously observed mortality (50%) of the CLCa KO mice was established to occur neonatally, and survivors displayed reduced body weight at this stage. Although the CLCb KO mice also displayed reduced body weight from this stage, it was less severe and the expected number of homozygous pups survived. CLCa KO mice displayed reduced fertility and uterine pyometra developed in ~40% of aged female CLCa KO mice (>4 mo), whereas CLCb KO

animals had no detectable uterine phenotype. The lumenal uterine epithelium in affected CLCa KO mice showed aberrant expression of the FOXA2 transcription factor. In a subset of mice developing pyometra, this epithelium displayed concomitant expression of K14 keratin, indicating a switch of epithelial phenotype (34, 39). The mice with pyometra also had a reduced number of uterine glands, but the glandular epithelium showed normal marker expression. Either CLCa or CLCb depletion before in vitro cyst formation resulted in aberrant lumen formation for both Ishikawa (uterine) and Caco-2 (gut) epithelial cell lines, supporting necessary roles for both CLC isoforms in lumen formation, with more severe defects observed for CLCa depletion. Interestingly, the differential distribution of CLCa compared with CLCb seen in cysts was not observed in mature uterine epithelium, suggesting either that the CLCs play differential roles depending on the pathway of lumen formation (38) or CLCs play differential roles at different stages of epithelial development. Together, these findings highlight a requirement for both CLCs in epithelial lumen development and support the previous suggestion from studies of neuronal phenotypes that CLCa sustains key clathrin functions that are fine-tuned by the expression of CLCb, such that their balanced function is required for clathrin pathways in different tissues (22).

In our previous analysis of the CLC KO mice, loss of each CLC isoform was shown to differentially affect synaptic vesicle recycling with CLCa KO mice displaying more severe neuronal defects (22). The perinatal mortality and reduced body weight of the CLCa KO mice reported here were also observed in mice lacking endophilin, another regulator of synaptic transmission with similar neurological abnormality (40). Thus, by analogy, the neurological defects could be the critical factor causing high neonatal mortality in CLCa KO mice through a variety of innervation pathways. Although neurological defects might indirectly affect fertility, reduced fertility of the CLCa KO mice could be related to the uterine lumen defects observed in the current study. Breeding the CLC KO mice in various combinations showed that CLCa is required for normal fertility. CLCa KO × CLCa KO breeding did not produce offspring, whereas CLCb KO × CLCb KO breeding was normal. In addition, CLCa HET × CLCa KO (male × female or female × male) breeding showed significantly reduced litter size, though CLCa WT × CLCa KO (male × female or female × male) breeding was unaffected. As such, the fertility defect cannot be entirely explained by disruption of uterine lumenal epithelia in the CLCa KO mice. More detailed measurements, such as fertilisation, embryo implantation, and embryo growth, would be needed to dissect the specific requirement for CLCa in breeding. For example trophoblast invasion in humans is accompanied by up-regulation of CLCb (20), so fertility could be sensitive or resilient to CLC expression in different combinations at multiple stages. Also, implantation involves embryo interaction with the epithelia lumen that is regulated by uterine gland function (41).

Defects in both epithelial cell identity and in gland number were observed in CLCa KO animals developing pyometra, which could be the extreme manifestation of uterine epithelial defects over time because pyometra appeared in aged mice (>4 mo). Pyometra occurs in middle- to older-aged small animals (42). In humans, although the incidence of pyometra is relatively rare (0.038–0.5% for gynecologic admissions), it is significantly increased in elderly patients (13.6%) (43, 44). The causes of pyometra are not entirely defined.

Hypothesized causes of pyometra which could be induced by CLC imbalance include age-related changes in epithelial barrier tissue caused by hormonal variation, leading to altered microbial infiltrate and subsequent inflammation (45). Regarding a role for microbial infiltrate, our investigation of epithelial organisation, proliferation, and polarity showed no obvious defect in the barrier properties of the uterine endometrium for the CLCa KO animals which did not have pyometra. Those with pyometra did show neutrophil infiltrate, a sign of infection, though mucin expression was not altered. However, because of limited tissue preparation from the affected animals it was not possible to assess thickness of the mucin layer or the presence of antibacterial peptides, which are also associated with uterine epithelial inflammation (46, 47). The ectopic expression of FOXA2 in the endometrium of the mice developing pyometra and their highly reduced gland number are indicators of endometrial epithelium defects, particularly because glands develop by invagination of the endometrial lumenal epithelium (41). Overexpression of FOXA2 has been shown to induce stratified epithelium in the mouse uterus (33). All the CLCa KO mice showed exogenous FOXA2 expression in their endometrial epithelium and only some pyometra-affected mice showed the K14 expression phenotype. This suggests that the CLCa KO animals all manifest an epithelial defect that can then be exacerbated by other factors, such as differential hormonal changes, that could induce more severe effects on endometrial identity and subsequent pyometra.

Hormonal changes are linked to pyometra (24, 42). Treatment of dogs with progestogens contributes to pyometra and pyometra generally develops during the secretory (luteal) phase of the oestrous cycle where the progesterone levels are elevated and endometrial secretions change, creating an environment that promotes microbiota growth (42, 48). Pyometra can also be induced in C57BL/6 mice by estradiol or bisphenol A (considered weak oestrogens) (24). The role of sex hormones in uterine epithelial changes and in the development of pyometra could be affected by CLCa loss through alterations in membrane traffic of the GPCRs that respond to these hormones, such as gonadotropin-releasing hormone receptor, follicle-stimulating hormone receptor, and luteinizing hormone receptor (49, 50). The identified role of CLCs in the regulation of GPCR trafficking (16, 17), and in lymphocyte receptor trafficking (9), could also have immunoregulatory effects, which have been linked to development of pyometra. Macrophage infiltration was seen in bisphenol A–treated mice and associated with susceptibility to pyometra (24). Furthermore, mice lacking the immunoregulatory cytokine meteorin-like protein also developed pyometra (51). Given that CLCa expression is dominant in immune cells and altered IgA antibody production in CLCa KO mice is associated with reduced TGF$\beta$R2 receptor uptake (9), immune cell defects in CLCa KO mice could also contribute to development of pyometra.

The ectopic expression of FOXA2 in the endometrium of CLCa KO mice and the further expression of K14 in these mice that develop pyometra indicates abnormality of the epithelial lumen. Although the K14 expression could result from FOXA2 expression, it is not clear whether these changes in lumenal cell identity are a response to defects in membrane traffic affecting epithelial development or a direct consequence of CLC depletion. Nonetheless, these findings

implicate CLCa function in development of the endometrial lumen. This epithelium, in turn, forms glandular epithelium through budding (41). The glandular epithelium appeared normal in our analysis, but gland number was reduced, presumably due to defects in the originating endometrial epithelium. Lumen formation by epithelial cysts in vitro depends on formation of the apical membrane initiation site between dividing cells, representing a different mechanism from formation of the uterine lumens (38). Although the CLCa and CLCb isoforms have different distributions relative to each other in tissue and cysts, they are both required for lumen formation, suggesting membrane traffic roles for both isoforms in establishing or maintaining polarity. CLCa is enriched at the apical side of the endometrial and glandular epithelia but is present throughout cyst epithelia. In the former, it is only loss of CLCa that affects function, whereas in the latter both CLCs are required for proper lumen formation. These findings demonstrate that CLC diversity is required for clathrin to function properly during membrane traffic regulating development of epithelia. The specialised but cooperative functions of the CLC isoforms in membrane traffic during epithelial formation mirrors their balanced role in synaptic vesicle traffic, and furthermore supports the view that CLCa plays the dominant roles in influencing physiological homeostasis with necessary regulation by CLCb. This regulation likely results from the fact that CLCs compete with each other for binding to clathrin heavy chain subunits (23) and contribute differential interaction with accessory proteins (21) and kinases, as well as different biophysical properties (22) to clathrin function.

# Materials and Methods

### Animals

CLCa or CLCb KO mice (both C57BL/6 background) were established as previously described (9, 22). All mice used in this study (including WT, heterozygotes and KOs used for breeding pairs in fertility studies) were produced by breeding *Clta*$^{-/+}$ or *Cltb*$^{-/+}$ heterozygotes. For fertility studies, individual breeding pairs of the indicated genotype were housed continuously in a breeding cage, and the litter size and frequency were monitored for 54-d or 4 mo as indicated. All animal procedures and breeding were conducted according to the Animals Scientific Procedures Act UK (1986) and in accordance with the ethical standards at University College London.

### Immunoblotting

Mouse tissue was harvested and snap-frozen in liquid nitrogen and stored at −80°C until further use. Tissue was homogenized in lysis buffer (50 mM Tris–HCl, pH 7.4, 150 mM NaCl, 1% NP-40, 0.5% sodium deoxycholate, 0.1% SDS, 1 mM EDTA) with complete EDTA-free proteinase inhibitor cocktail (Roche) and then incubated on ice for 1 h. After centrifugation at 10,000*g*, 4°C, 10 min, the supernatant was taken and the protein concentration was determined by BCA assay (Thermo Fisher Scientific). Proteins were separated by

SDS–PAGE gel and transferred to Protran nitrocellulose membrane (0.2 *μ*m; GE Healthcare). Immunoblotting was performed by incubating with primary antibodies diluted in TBS (20 mM Tris, pH 7.6, 150 mM NaCl) containing 2% BSA overnight at 4°C. Primary antibodies used were anti-CLC (CON.1, made in house) (26) and anti-tubulin (1:10,000; Abcam). Blots were incubated with IRDye800/700-conjugated secondary antibodies (1:5,000; LI-COR Biosciences), 1 h, room temperature. Proteins were detected with LI-COR Odyssey Imager (LI-COR Biosciences) and analysed using Image Studio Lite 5.2 software (LI-COR Biosciences). Immunoblotting of lysate from cultured cells was performed using the same protocol.

### Histology and immunohistochemistry

Mouse uteri were removed and fixed in 4% PFA (pH 7.4; Sigma-Aldrich) at 4°C overnight. Specimens were dehydrated through graded ethanol solutions, paraffin-embedded, and cut into 5 *μ*m sections. Sections were then deparaffinized and hydrated through graded ethanol solutions. For antigen retrieval, sections were incubated in sodium citrate buffer (10 mM sodium citrate, 0.05% Tween 20, pH 6.0), EDTA buffer (1 mM EDTA, 0.05% Tween 20, pH 8), or Tris-EDTA buffer (10 mM Tris base, 1 mM EDTA, 0.05% Tween 20, pH 9.0) for 20 min at 95°C. Slides were incubated with blocking buffer 1 (2% BSA, PBS) for 15 min and blocking buffer 2 (1% BSA, 0.3% Triton, PBS) for 15 min before incubation with primary antibodies diluted in blocking buffer 2 at 4°C overnight. Primary antibodies and antigen retrieval method used were anti-CLCa (1:200, citrate buffer; Sigma-Aldrich), anti-CLCb (1:200, citrate buffer; Sigma-Aldrich), anti-FOXA2 (1:100, EDTA buffer; Cell Signaling), anti-ZO-1 (1:100, citrate buffer; Thermo Fisher Scientific), E-cadherin (1:100, citrate buffer; BD), Ki-67 (1:200, EDTA buffer; Cell Signaling), anti-SOX9 (1:200, citrate buffer; Cell Signaling), anti-K14 (1:100, citrate buffer; Abcam), anti-mucin 1 (1:200, Tris–EDTA buffer; Abcam) anti-Ly-6G (1A8)-Alexa 647 (1:50, citrate buffer; BioLegend), *β*-catenin Alexa 488 (1:50, EDTA; BD). Slides were washed with PBST (0.1% Tween 20; PBS) and incubated with Alexa Fluor–conjugated secondary antibody (1:500; Thermo Fisher Scientific) and DAPI (1 *μ*g/ml; Thermo Fisher Scientific) for 1 h. After PBST washes, slides were mounted with ProLong Diamond antifade media (Thermo Fisher Scientific). Images were obtained using a Leica TCS SP8 inverted laser–scanning confocal microscope with a PLAN APO 40X 1.3NA oil immersion objective, using four laser lines. Dyes were sequentially excited at 405 nm (DAPI), 488 nm (Alexa Fluor 488), 543 nm (Alexa Fluor 555), and 633 nm (Alexa Fluor 647). 1,024 × 1,024 pixel images where acquired using a photomultiplier detector, with the following emission wavelengths: DAPI: 410–470 nm; Alexa Fluor 488: 500–545 nm; Alexa Fluor 555: 560–610 nm; Alexa Fluor 647: 645–695 nm. Images were captured using Leica LAS X software and subsequently analysed using Image J.

### Cell lines

Ishikawa cells (ECACC) were grown in minimum essential medium (Sigma-Aldrich) supplemented with 5% FBS (Thermo Fisher Scientific), 1% non-essential amino acids (Sigma-Aldrich), 2 mM L-glutamine (Thermo Fisher Scientific), and 100 U/ml penicillin/100 *μ*g/ml streptomycin (P/S) (Thermo Fisher Scientific). Caco-2

cells were grown in DMEM (Thermo Fisher Scientific) supplemented with 10% FBS, 2 mM L-glutamine and P/S.

### Cyst formation and siRNA treatment

For cyst formation, Ishikawa and Caco-2 cell lines were transfected with 20 nM siRNA (QIAGEN) using jetPRIME (Ishikawa) or INTERFERin (Caco-2) (both Polyplus Transfection) transfection regent following manufacturer's instructions. Target sequences for siRNA used were CLTA: 5′-AA AGA CAG TTA TGC AGC TAT T-3′, CLTB: 5′-AAG GAA CCA GCG CCA GAG TGA-3′, CLTC: 5′-AAG CAA TGA GCT GTT TGA AGA. 72 h after transfection, cells were trypsinised into single cells. 6,000 cells were mixed with 30% Matrigel (Corning) in PBS and plated in a well of an eight-well EZ slide (Millipore) which had been pre-coated with 5 μl Matrigel. Cells were incubated in a 37°C, 5% $CO_2$ incubator for 30 min to allow Matrigel to set before then being supplemented with 400 μl culture medium. Culture medium was changed every 2 d. Cholera toxin (0.1 μg/ml; Sigma-Aldrich) was added 24 h prior fixation where indicated.

### Immunofluorescence staining

Cysts were washed twice with PBS and fixed in 4% PFA at room temperature for 1 h before unreacted PFA was quenched with glycine. After PBS washes, cysts were permeabilized with 0.1% Triton X-100 in PBS for 30 min, followed by blocking with blocking buffer 1 for 1 h. Cysts were then incubated with primary antibodies diluted in blocking buffer overnight at 4°C. Primary antibodies used were anti-CLCa (1:250; Sigma-Aldrich), anti-CLCb (1:250, LCB.1, made in house ([23])), ZO-1 (1:100; Thermo Fisher Scientific), E-cadherin (1:100; BD). After PBS washes, cysts were incubated with Alexa Fluor 488 or 647 conjugated–secondary antibody (1:500; Thermo Fisher Scientific) and Alexa 546–conjugated phalloidin (1:50; Thermo Fisher Scientific) diluted in blocking buffer for 1.5 h at room temperature. Cells were washed with PBS and stained using 1 μg/ml DAPI in PBS (Thermo Fisher Scientific) before slides were mounted with ProLong Diamond antifade media (Thermo Fisher Scientific). Images were obtained using a Leica TCS SP8 inverted laser-scanning confocal microscope with a HC PLAN APO 63X 1.40 NA CS2 oil immersion objective, using four laser lines. Dyes were sequentially excited at 405 nm (DAPI), 488 nm (Alexa Fluor 488), 543 nm (Alexa Fluor 546), and 633 nm (Alexa Fluor 647). 1,024 × 1,024 pixel images where acquired using a photomultiplier detector, with the following emission wavelengths: DAPI: 410–470 nm; Alexa Fluor 488: 500–545 nm; Alexa Fluor 546: 565–595 nm; Alexa Fluor 647: 645–695 nm. Images were captured using Leica LAS X software and subsequently analysed using Image J.

### Transient transfection of Flag–FOXA2

For transient DNA transfection, Ishikawa cells were seeded at 70,000 cells/well of a six well dish. The following day, cells were transfected with 0.75 μg Flag–FOXA2 (number 153110; Addgene) using jetPRIME transfection reagent (Polyplus Transfection), following manufacturer's instructions. Control samples were mock transfected using transfection reagent without DNA. 24 h post transfection, cells were collected for immunoblotting or mRNA extraction.

### mRNA extraction, cDNA synthesis, and qPCR

mRNA was collected from cells in six well dishes using TRIzol reagent (Invitrogen) following manufacturer's instructions. Briefly, cells were lysed and homogenised using 1 ml TRIzol reagent per well. Lysates were transferred to a microfuge tube and incubated for 5 min before adding 0.2 ml chloroform and mixing. After incubating for 3 min, samples were centrifuged at 12,000$g$ for 15 min at 4°C. The aqueous phase was subsequently collected and RNA precipitated by incubation with 0.5 ml isopropanol, 10 min, 4°C. Samples were centrifuged at 12,000$g$ for 10 min at 4°C, and then pellets washed in 1 ml 70% ethanol before a final centrifugation at 7,500$g$, 5 min, 4°C. The supernatant was discarded and pellet air dried for 5 min before resuspending in 30 μl RNase-free water. cDNA was synthesised from the isolated mRNA using the LunaScript RT SuperMix Kit (New England Biolabs) following manufacturer's instructions. qPCR was performed using a Chromo4 PTC-200 Real-Time PCR Detector system (Bio-Rad) with SYBR-Green JumpStart Taq ready mix (Merck). PCR conditions were 94°C for 2 min, followed by 40 three-step cycles of 94°C for 15 s, 60°C for 30 s, and 72°C for 30 s. GAPDH was used as the endogenous gene controls. Melt curves were used to determine that a single PCR product was produced by each primer pair. Each qPCR was performed in triplicate with the following primers: *AREG* (forward: GTGGTGCTGTCGCTCTTGATA; reverse: CCCCAGAAAATGGTTCACGCT), *CTNNB1* (forward: CACAAGCAGAGTGCTGAAGGTG, reverse: GATTCCTGAGAGTCCAAAGACAG), *C3* (forward: GTGGAAATCCGAGCCGTTCTCT, reverse: GATGGTTACGGTCTGCTGGTGA), *FOXA2* (forward: CCCCACAAAATGGACCTCAAG, reverse: GAGTACACCCCCTGGTAGTAG), *GAPDH* (), *IHH* (forward: AACTCGCTGGCTATCTCGGT, reverse: GCCCTCATAATGCAGGGACT), *KRT14* (forward: TGAGCCGCATTCTGAACGAG, reverse: GATGACTGCGATCCAGAGGA), *LIF* (forward: CTGTTGGTTCTGCACTGGAA, reverse: CCCCTGGGCTGTGTAATAGA), *LTF* (forward: ATGCTGGAGATGTGGCTT, reverse: CCTTTCGGCTTTATTTGGT), *MUC1* (forward: TGCCGCCGAAAGAACTACG, reverse: TGGGGTACTCGCTCATAGGAT), *RNF43* (forward: AAAATCCAGCCTCTCTGCCC, reverse: ACAACCACACTGGCTGTGAA), *SPINK1* (forward: TGACTCCCTGGGAAGAGAGG, reverse: AGTCCCACAGACAGGGTCAT), *Ltf* (forward: ATCTCTGTGCCCTGT, reverse), *Muc1* (forward: TTCCAACCCAGGACACCTAC, reverse: ATTACCTGCCGAAACCTCCT).

### Statistical analysis

Statistical analysis was performed using Prism 7 (GraphPad). Methods of comparison are stated in figure legends. Mean ± SE is shown.

# Supplementary Information

# Acknowledgements

We acknowledge University College London IQPath, University College London Institute of Neurology, for processing tissue slices and for H&E staining, University College London KLB animal facility for mouse maintenance and William Andrews of the University College London Biosciences Molecular

Biology Facility for assistance with qPCR. This work was supported by Wellcome Trust Grant 107858/Z/15/Z and MRC grant MR/S008144/1 (both to FM Brodsky).

## Author Contributions

Y Chen: conceptualization, formal analysis, investigation, visualization, methodology, and writing—original draft, review, and editing.
K Briant: conceptualization, formal analysis, investigation, visualization, methodology, project administration, and writing—original draft, review, and editing.
MD Camus: formal analysis, investigation, and project administration.
FM Brodsky: conceptualization, supervision, funding acquisition, project administration, and writing—original draft, review, and editing.

## Conflict of Interest Statement

The authors declare that they have no conflict of interest.

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
