## [Reviewer comments · Life Science Alliance]

Life Science Alliance

Clathrin light chains CLCa and CLCb have non-redundant roles in epithelial lumen formation

Yu Chen, Kit Briant, Marine Camus, and Frances Brodsky

DOI: <https://doi.org/10.26508/lsa.202302175>

Corresponding author(s): Frances Brodsky, University College London

Review Timeline:

Submission Date:	2023-05-19
Editorial Decision:	2023-06-26
Revision Received:	2023-09-18
Editorial Decision:	2023-10-03
Revision Received:	2023-10-14
Accepted:	2023-10-16

Transaction Report:

June 26, 2023

Re: Life Science Alliance manuscript #LSA-2023-02175

Prof. Frances M Brodsky
University College London
Division of Biosciences
Gower Street
London WC1E 6BT
United Kingdom

Dear Dr. Brodsky,

Thank you for submitting your manuscript entitled "Clathrin light chains CLCa and CLCb have non-redundant roles in epithelial lumen formation" to Life Science Alliance. The manuscript was assessed by expert reviewers, whose comments are appended to this letter. We invite you to submit a revised manuscript addressing the Reviewer comments.

Thank you for this interesting contribution to Life Science Alliance. We are looking forward to receiving your revised manuscript.

Sincerely,

B. MANUSCRIPT ORGANIZATION AND FORMATTING:

Reviewer #1 (Comments to the Authors (Required)):

Clathrin-mediated endocytosis regulates many aspects of cell physiology, such as by control of nutrient uptake, receptor signaling, and control of cell surface abundance of various proteins. A fundamental question in the field has been how a common mechanism involving clathrin and its interacting proteins can control such diverse functions, and exhibit context-specific regulation of various receptors and signals, including tissue-specific regulation. Many studies have shown that in addition to a core requirement for clathrin heavy chain and the adaptor protein AP2 that are "core" structural components of clathrin-coated pits, other proteins function to regulate and adapt this process for receptor- (cargo) and tissue-specific functions. This study examines how clathrin light chain a and b (CLCa and CLCb) function in regulation of tissue development and function, focusing on the development of the uterus.

Building on previous work that established CLCa and CLCb KO animals, this study first finds that CLCa $-/-$ animals are born with expected ratios, but undergo loss of viability 3 days after birth. Surviving CLCa $-/-$ mice as well as CLCb $-/-$ mice exhibit a reduction of body weight. Female CLCa $-/-$ or $+/-$ mice exhibited reduced litter size when mated with CLCa $+/-$ or $+/-$ males, but exhibited normal litter sizes when mated with CLCa $+/+$ males. CLCa $-/-$ mice exhibited loss of glandular structures in the uterus, and 42% of CLCa $-/-$ mice exhibited increased neutrophil invasion (Ly6G-positive) into an enlarged uterus filled with a cloudy uterus, consistent with pyometra. CLCa and CLCb were both enriched in the luminal and glandular epithelium of the uterus, and both were concentrated at the apical domain of these cells. Loss of CLCa led to increased expression of FOXA2 (a transcription factor that regulates epithelial development) in the luminal epithelial cells; however, this did not appear to impact proliferation of epithelial cells measured by Ki67 or SOX9 expression. In CLCa $-/-$ animals exhibiting pyometra, but not CLCa $-/-$ mice not exhibiting pyometra, there was increased keratin 14 expression, indicative of increased abundance of stratified squamous epithelial cells, suggestive of some differentiation defects in the luminal epithelia in the uterus of CLCa $-/-$ mice, even though no defects were observed in CLCa $-/-$ in terms of cell adhesions (E-cadherins and ZO-1) or mucin-1 expression.

To further probe the mechanism by which CLCa or CLCb may affect epithelial cell differentiation and formation of an epithelial monolayer in the uterus, a cell culture model was next examined, using Ishikawa cells (human endometrial adenocarcinoma cells) and Caco-2 cells (human colorectal adenocarcinoma cells), which when grown in Matrigel for 4-6 days exhibit cyst formation. Loss of either CLCa, CLCb, or clathrin heavy chain (CHC) by siRNA gene silencing led to impairment of cyst lumen formation. This suggests that perturbation of CLCa/b or CHC leads to perturbation of apical-basolateral polarity in these epithelial cell models.

Overall, this study provides important new insights into how clathrin-mediated endocytosis may be uniquely regulated by the proteins that interact with clathrin heavy chain such as CLCa and CLCb, to specify tissue- and cell-specific control of cell physiology. The experiments in this study are well designed, and the combination of study of knockout animals and 3D cell culture models with manipulation of CLCa and b allow strong conclusions to be made about the roles of CLCa/b in epithelial differentiation and development. It is clear that more research can be done to delineate specific endocytosis defects in epithelial cells in the uterus or in cell culture models upon loss of function of CLCa/b, such as identification of receptors that have endocytosis selectively impacted by CLCa and/or CLCb. However, these experiments are beyond the scope of the current study and this current manuscript presents interesting results on the field of endocytosis may build. The results of this study are novel and will be of interest broadly to cell biologists as well as to researchers studying fertility and uterine development and physiology. Some comments for the authors to consider are indicated below.

Major comments:

- 1) The model presented has female mice exhibiting development defects in the uterus as a result of loss of CLCa, either hetero- or homozygously. It is thus not clear how this can explain that mating of a CLCa $-/-$ female with a CLCa $+/+$ male exhibits no litter size defects. The latter would imply that there is significant contribution of the defects in litter size and fertility following fertilization. It would be useful to discuss this result in the context of the model of uterus development defects presented in this manuscript. This can likely be addressed by minor modifications to the text without requiring additional experiments that would be out of scope for the current study.
- 2) For the cell culture experiments (e.g. Figure 7), it would be useful to have a western blot showing silencing of CLCa and CLCb

similar to that showing the loss of CLCa and CLCb in KO animals (figure 4A).

3) For the images shown in Figure 7D-E, it would be useful to consider adding some scoring of the cysts that show apical polarization of ZO1 and E-Cadherin. These images show clearly that there is disruption of polarization upon silencing of CLCs or CHC, but showing the percentage of cells that show disruption of apical polarization will be useful to understand the penetrance of this phenotype.

Minor comments:

- 1) On page 12 (results section), regarding the statement "However, the cellular distribution in CLC-depleted Ishikawa cysts was not as organised as in the control-treated cysts...": it would be helpful to be more explicit about what is meant by the "cellular distribution".
- 2) On page 13 (results section), regarding the statement "while treatment with siRNA targeting CLCb increased the proportion of cysts with multiple lumens, although this was not statistically significant (Fig. 7I-J)...": apparent differences between conditions should not be referred to as different if they are not statistically significantly different.
- 3) The methods are largely well described. However, for the "Immunofluorescence staining" section, the laser illumination and emission filter conditions for the confocal microscope used should be indicated.
- 4) On page 6 of the Results, regarding the statement: "Loss of CLCa had a greater effect on pup body weight than loss of CLCb, with an average reduction of 21.52% for CLCa KO animals relative to WT controls at PN7, compared to an average 19.1% weight reduction for CLCb KO pups relative to WT". It may be prudent not to make conclusions about CLCa having a greater effect on bodyweight than CLCb unless there is statistical analysis to support this, showing that the difference in body weight in CLCa KO vs WT is significantly more than the difference in body weight than CLCb KO vs WT.

Reviewer #2 (Comments to the Authors (Required)):

This is a somewhat descriptive study that continues to analyze the role of CLCa and CLCb during mice development in vivo. The study is not really mechanistic and essentially leads to conclusion that CLCa and CLCb has somewhat differential roles, but also are likely compensated by other trafficking mechanisms, thus, leading to very mild defects in epithelia organization and polarization (unlike in tissue culture models). Nevertheless, despite its limited innovation and descriptive nature, it would be usefully to publish it since we still know little about differences in CLCa and CLCb function in vivo and all too often (as also demonstrated in this study) in vivo phenotypes do not recapitulate the observations in vitro. The study also has few technical issues (see below) that should be addressed before its suitable for publication.

- 1) Does ectopic FOXA2 expression leads to ectopic (in epithelia) expression of FOXA2 target genes? That is relatively easy to test and should be included in this manuscript.
- 2) In figure 7 authors need to show the efficiency of CLCa and CLCb knock-down. That is especially relevant in 3D assays since cells are incubated 4-6 days after 72 hour knock-down. During that time, it is quite likely that siRNA has been used up and protein level are rebounding. Consequently, western blot analysis need to be done at 72 hours (before embedding cells in Matrigel) and then after another 4-6 days. I realize that once cannot do western from Matrigel-embedded cells, but western at least should be done after 4-6 day growth on 2D plates.
- 3) siRNA are notorious for off-target effect. All knock-down experiments need to be done using at least two different siRNAs.

Minor:

- 1) In Figure 1C Y-axis values should start at 0 rather than 0.8.

Reviewer #3 (Comments to the Authors (Required)):

Clathrin is a protein that is essential for many membrane trafficking events. The major clathrin gene in eukaryotic cells is CHC17 which is composed of clathrin heavy chain and three associated light chains. The light chains arise from two independent genes, CLCa and CLCb, which can undergo alternative splicing to yield neuronal and non neuronal forms. There is a high degree of conservation between CLCa and CLCb with ~60% sequence identity. Until fairly recently the precise roles of the CLCs in membrane trafficking was relatively opaque but recent studies have begun to elucidate functional roles for both light chains which suggest both independent and overlapping functions. The Brodsky lab generated CLCa and CLCb knockout mice which have provided important insight into the differing functions of CLCa and CLCb in immune and neuronal cells. This paper extends this work by investigating the roles of the light chains in uterine epithelia. Specifically, the manuscript demonstrates that while CLCb KO mice have no issues with survival or fertility, CLCa mice have significant neonatal mortality as well as reduced fertility. Notably, >40% of aged females developed uterine pyometra.

The main content of the manuscript is an exploration of the underlying cause of the latter phenotype. The authors show that wild-type mice have a slightly increased ratio of CLCa to CLCb in uterine epithelia. In the absence of CLCa the authors point to a slight increase in CLCb expression.

Minor point: looking at the blot it seems like the increase might be quite significant and is probably worth quantifying since this supports the authors premise that loss of CLCa cannot be compensated for by increased CLCb expression.

The authors then explore expression of FOXA2, a transcription factor required for epithelial development and observed aberrant expression in CLCa KO mice which was significantly worse in those with pyometra. Through analysis of SOX9 and Ki67 expression, the authors could eliminate effects on cell proliferation and instead their experiments implicated stratification of the epithelium because of increased expression of keratin14 in affected mice. Finally the authors investigate the role of CLCs in epithelia cyst formation in vitro using Ishikawa and Caco2 cysts, where they show disruption of epithelial polarity although to different extents in the different cell lines. Together the data in the paper support a key requirement for CLCa in uterine epithelium physiology with CLCb playing a regulatory role.

The Holy Grail of defining membrane trafficking pathways is to understand how they contribute to organismal physiology and this paper makes a significant contribution in this area. The experiments have been well-executed and the manuscript is clearly written. I recommend that it be published as is.

18 September 2023

With this letter, we submit a revised version of our manuscript by Chen, Briant et al entitled "Clathrin light chains CLCa and CLCb have non-redundant roles in epithelial lumen formation" to be considered for publication in Life Science Alliance. We append a point-by-point response to the reviews and hope that the manuscript is now found acceptable.

Sincerely yours,

Frances M Brodsky, DPhil, FMedSci

Point by point response to the reviews of manuscript #LSA-2023-02175 “Clathrin light chains CLCa and CLCb have non-redundant roles in epithelial lumen formation”.

We appreciate the constructive criticism from the reviewers and are pleased that the study was seen to provide “important new insights into how clathrin-mediated endocytosis may be uniquely regulated by the proteins that interact with clathrin heavy chain such as CLCa and CLCb, to specify tissue- and cell-specific control of cell physiology” (Reviewer #1) and that this study “makes a significant contribution” to understanding how “membrane traffic pathways contribute...to organismal physiology” (Reviewer #3). We appreciate also that Reviewer #2 found the study “useful... to publish since we still know little about differences in CLCa and CLCb function in vivo and all too often (as also demonstrated in this study) in vivo phenotypes do not recapitulate the observations in vitro.” Reviewer #3 had no editorial suggestions and recommended “that it be published as is”. Below we address the comments of the other two Reviewers, with their suggestions in italics and our reply in normal font.

Reviewer #1

Major comments:

1) The model presented has female mice exhibiting development defects in the uterus as a result of loss of CLCa, either hetero- or -homozygously. It is thus not clear how this can explain that mating of a CLCa -/- female with a CLCa +/- male exhibits no litter size defects. The latter would imply that there is significant contribution of the defects in litter size and fertility following fertilization. It would be useful to discuss this result in the context of the model of uterus development defects presented in this manuscript. This can likely be addressed by minor modifications to the text without requiring additional experiments that would be out of scope for the current study.

This is an interesting point, and one that we have pondered, without successful explanation. It is correct that the uterine phenotype of the CLCa KO mice cannot fully explain the fertility defects noted, since these mice can breed successfully with WT males and produce normal-sized litters, while their breeding with heterozygous males is partially impaired. As the reviewer noted, finding an explanation for these phenotypes would require considerably more experimentation, beyond the scope of the present study, as several stages of fertility would need to be examined. We have now altered the text of the Discussion section (page 16) to deal with this point in more detail and to mention post-fertilization stages where breeding defects could manifest due to changes in clathrin function.

2) For the cell culture experiments (e.g. Figure 7), it would be useful to have a western blot showing silencing of CLCa and CLCb similar to that showing the loss of CLCa and CLCb in KO animals (figure 4A).

We apologise for the omission of data confirming that clathrin components remain depleted at the stage of cyst phenotype analysis that was requested by both Reviewer #1 and Reviewer #2, and agree this data should have been included. We have now added the relevant data as supplementary figure S5, described and cross-referenced in the text on page 12&13. We note that for depletion of clathrin components from cysts, the efficiency of siRNA transfection was better for Ishikawa cysts than for Caco-2 cysts. The former can easily be detected by blotting the whole culture, 9 days after knockdown (Fig. S5A & B), while for the latter knockdown in cultured cysts was more variable, with less efficient depletion shown by blotting at Day 0 (Fig.S5 E). However, we observed that cysts that originated from a Caco-2 cell that had been efficiently depleted of the siRNA target after 72h (cyst growth day 0) remained efficiently depleted at day 6 as determined by IF (Fig. S5 C-D). Thus, protein levels in the cysts do not recover for the period of cyst growth.

3) For the images shown in Figure 7D-E, it would be useful to consider adding some scoring of the cysts that show apical polarization of ZO1 and E-Cadherin. These images show clearly that there is disruption of polarization upon silencing of CLCs or CHC, but showing the percentage of cells that show disruption of apical polarization will be useful to understand the penetrance of this phenotype.

We appreciate the reasoning for the request to quantify the phenotypes presented in figure 7C-E. We did quantify apico-basal polarity disruption utilising the lumenogenesis assay in Fig 7H-J. However, it is clear that we did not explain sufficiently why this assay quantifies the polarity disruption observed in Fig 7C-E. During in vitro cyst development, normal lumen formation is directly tied to the establishment of apico-basal polarity, during which an apical proteome accumulates at an apical membrane initiation site (AMIS) that originates during cell division. This AMIS forms the site for future lumen growth. Thus, disruption of the formation of an AMIS will lead to disrupted lumen formation. As such, we consider lumen formation (using phalloidin to visualise lumens) a better assay for the general organisation of apical polarity in the cysts than the localisation of individual markers. We have attempted to clarify this in the text (page 13&14) and we have now included, as supplementary figure S6, examples of co-staining for phalloidin, ZO-1 and E-cadherin to demonstrate the overlap between ZO-1 and apical phalloidin rings in control cysts, and that loss of single lumens coincides with mislocalisation of ZO-1.

Minor comments:

1) *On page 12 (results section), regarding the statement "However, the cellular distribution in CLC-depleted Ishikawa cysts was not as organised as in the control-treated cysts...": it would be helpful to be more explicit about what is meant by the "cellular distribution".*

We realise that this phrasing was ambiguous. We were referring to the organisation of cells within the cyst. In a normal cyst the cells are organised in a symmetrical ring, which becomes more disorganised on clathrin depletion. This change in organisation was detected by DAPI staining, and we have now clarified our comment and described the detection method in the text on page 13.

2) *On page 13 (results section), regarding the statement "while treatment with siRNA targeting CLCb increased the proportion of cysts with multiple lumens, although this was not statistically significant (Fig. 7I-J)...": apparent differences between conditions should not be referred to as different if they are not statistically significantly different.*

We have removed the reference to this result from the text (now page 14).

3) *The methods are largely well described. However, for the "Immunofluorescence staining" section, the laser illumination and emission filter conditions for the confocal microscope used should be indicated.*

These details have now been added to the methods in two sections where immunofluorescence was used on pages 21-23.

4) *On page 6 of the Results, regarding the statement: "Loss of CLCa had a greater effect on pup body weight than loss of CLCb, with an average reduction of 21.52% for CLCa KO animals relative to WT controls at PN7, compared to an average 19.1% weight reduction for CLCb KO pups relative to WT". It may be prudent not to make conclusions about CLCa having a greater effect on bodyweight than CLCb unless there is statistical analysis to support this, showing that the difference in body weight in CLCa KO vs WT is significantly more than the difference in body weight than CLCb KO vs WT.*

We have removed the reference to this from the text (page 6).

Reviewer #2

Major comments:

1) Does ectopic FOXA2 expression leads to ectopic (in epithelia) expression of FOXA2 target genes? That is relatively easy to test and should be included in this manuscript.

This is an interesting question that we had addressed by tissue analysis prior to submission (not originally included in the manuscript) and then followed up during the revision period with new experimentation. We further note that earlier published studies (not from our group) had already identified some affected target genes in transgenic mice aberrantly expressing FOXA2 in the uterus (refs 33 and 35 in manuscript). Initially, we had analysed transcript levels from the uterine tissue of the CLC mutant mice, including the FOXA2 target genes *Ltf* and *Muc1*, but found that effects on their expression was not consistent between individual animals. We now include this data in a new supplementary Figure S3A and mention it on page 11. However, from this earlier study (by necessity) analysed RNA extracted from whole organ tissue, and not specifically from epithelia. To further assess the effect of aberrant FOXA2 expression in epithelia, we then ectopically expressed Flag-FOXA2 in the endometrial epithelial Ishikawa cell line and looked for expression of reported uterine FOXA2 target genes. This recent experiment confirmed that ectopic FOXA2 expression in Ishikawa cells, which do not express endogenous FOXA2, did significantly induce expression of the uterine FOXA2 target gene *SPINK1*. This new data has been added as new supplementary figures S3B-E and is referred to in the text on page 11. Additional methodology has been added on pages 23-25.

2) In figure 7 authors need to show the efficiency of CLCa and CLCb knock-down. That is especially relevant in 3D assays since cells are incubated 4-6 days after 72 hour knock-down. During that time, it is quite likely that siRNA has been used up and protein level are rebounding. Consequently, western blot analysis need to be done at 72 hours (before embedding cells in Matrigel) and than after another 4-6 days. I realize that once cannot do western from Matrigel-embedded cells, but wester at least should be done after 4-6 day growth on 2D plates.

As described in the response to reviewer 1 above, we have added the requested evidence of the siRNA knockdown efficiency in supplementary figure S5. This data is referred to in the text on page 12/13.

3) siRNA are notorious for off-target effect. All knock-down experiments need to be done using at least two different siRNAs.

While we agree that off-target effects can be a potential problem with siRNA experiments, we believe that is not an issue in this case. This is because the three siRNAs used here (siCLCa, siCLCb and siCHC17), which target separate components of clathrin, produce similar phenotypes. Given that these siRNAs have completely independent sequences, the off-target effects, if any, would be different for each siRNA and it is therefore unlikely that the off-target effects of each siRNA would phenocopy each other. Therefore, we believe in this instance it is reasonable to ascribe the effects of these siRNAs to depletion of clathrin components. Additionally, these siRNA sequences have been used for almost 20 years by our laboratory (e.g. Vassilopoulos et al, 2009, Science; Majeed et al, 2014, Nat Comm) and others (e.g. Huang et al, 2004, JBC) with verified phenotypes related to clathrin function. In the published experiments from our laboratory, we have rescued the phenotypes by re-transfection of the target gene encoding CLCa, CLCb or CHC17.

Minor comments:

1) In Figure 1C Y-axis values should start at 0 rather than 0.8.

This has been changed.

October 3, 2023

RE: Life Science Alliance Manuscript #LSA-2023-02175R

Prof. Frances M Brodsky
University College London
Division of Biosciences
Gower Street
London WC1E 6BT
United Kingdom

Dear Dr. Brodsky,

Thank you for submitting your revised manuscript entitled "Clathrin light chains CLCa and CLCb have non-redundant roles in epithelial lumen formation". We would be happy to publish your paper in Life Science Alliance pending final revisions necessary to meet our formatting guidelines.

- please add ORCID ID for the corresponding author - you should have received instructions on how to do so
- please place the figure legends after the Reference section
- please add callouts for Figures 4D and S5A, C to your main manuscript text

A. FINAL FILES:

B. MANUSCRIPT ORGANIZATION AND FORMATTING:

Sincerely,

Reviewer #1 (Comments to the Authors (Required)):

The revised manuscript addresses all the points raised during review of the initial submission. I have no further comments. This is an interesting and informative study that will be of significant interest.

October 16, 2023

RE: Life Science Alliance Manuscript #LSA-2023-02175RR

Prof. Frances M Brodsky
University College London
Division of Biosciences
Gower Street
London WC1E 6BT
United Kingdom

Dear Dr. Brodsky,

Thank you for submitting your Research Article entitled "Clathrin light chains CLCa and CLCb have non-redundant roles in epithelial lumen formation". It is a pleasure to let you know that your manuscript is now accepted for publication in Life Science Alliance. Congratulations on this interesting work.

DISTRIBUTION OF MATERIALS:

Again, congratulations on a very nice paper. I hope you found the review process to be constructive and are pleased with how the manuscript was handled editorially. We look forward to future exciting submissions from your lab.

Sincerely,
